# LITEFORMER: LIGHTWEIGHT EVOFORMER FOR PROTEIN STRUCTURE PREDICTION

## ABSTRACT

AlphaFold2 has achieved seminal success in predicting structures from amino acid sequences with remarkable atomic accuracy. However, its Evoformer module faces a critical challenge in terms of high memory consumption, particularly concerning the computational complexity associated with sequence length $L$ and the number of Multiple Sequence Alignments (MSA), denoted as $s$. This challenge arises from the attention mechanism involving third-order MSA and pair-wise tensors, leading to a complexity of $\mathcal{O}(L^3 + sL^2)$. This memory bottleneck poses difficulties when working with lengthy protein sequences. To tackle this problem, we introduce a novel and lightweight variant of Evoformer named Liteformer. Liteformer employs an innovative attention linearization mechanism, reducing complexity to $\mathcal{O}(L^2 + sL)$ through the implementation of a bias-aware flow attention mechanism, which seamlessly integrates MSA sequences and pair-wise information. Our extensive experiments, conducted on both monomeric and multimeric benchmark datasets, showcase the efficiency gains of our framework. Specifically, compared with Evoformer, Liteformer achieves up to a 44% reduction in memory usage and a 23% acceleration in training speed, all while maintaining competitive accuracy in protein structure prediction.

## 1 INTRODUCTION

Recently, computational methods (Roy et al., 2010; Brini et al., 2020; Baek et al., 2021; Jumper et al., 2021) based on the evolutionary history of proteins have shown the immense capability of predicting three-dimensional protein structure from sequence. A significant milestone was achieved with DeepMind's AlphaFold2 (Jumper et al., 2021) in 2021. This groundbreaking model derives protein structures by leveraging evolutionary history and pair-wise evolutionary correlations. Incorporating advanced deep learning models, such as transformer-style models, and accumulating a large database of protein sequences and structures, AlphaFold2 is capable of predicting protein structure with atomic accuracy.

Specifically, the AlphaFold2 pipeline first utilizes a stack of transformer blocks in Evoformer to update evolutionary information within multiple sequence alignments (MSA) representation and pair-wise correlation representation. For MSA representation updates, row-wise biased attention is employed to integrate pair-wise information. And pair representation update operates both column-wise and row-wise with triangle bias to satisfy the triangle inequality of distances. Subsequently, the updated information is passed to the structure module, which iteratively reconstructs a three-dimensional structure of the primary sequence.

However, since both the row and column dimensions of the pair representation, along with the row dimension of the MSA representation, are the same as the primary sequence length $L$. And the column dimension of the MSA representation $s$ is also of a comparable magnitude to the primary sequence length [1], coupled with the quadratic complexity introduced by canonical attention mechanisms, these update operations ultimately yield $\mathcal{O}(L^3 + sL^2)$ memory and computational complexity. Here the analysis excludes the head dimension $d$ due to its relatively small magnitude.

We observe that the transformer blocks within Evoformer constitute the major portion of memory usage in the overall model. This limitation hinders AlphaFold2 from scaling up to a larger model and

---

[1]The distribution of sequence length and MSA number can be seen in Appendix 7.

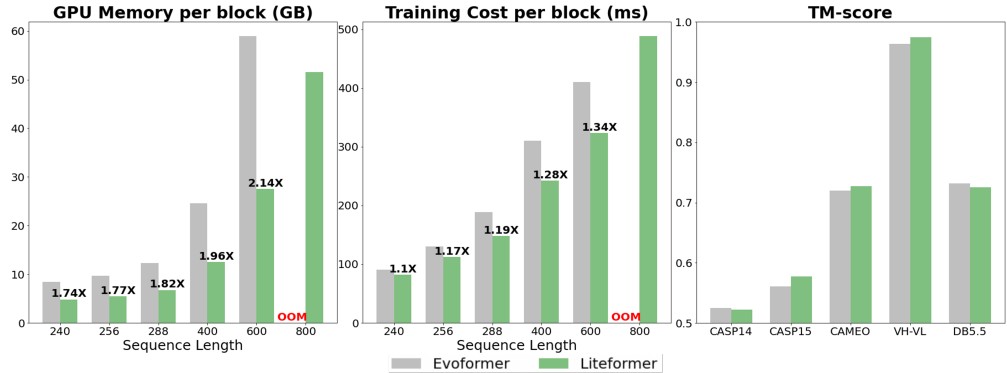

Figure 1: Comparison of our Liteformer with Evoformer in terms of training computational costs (GPU memory per block (left) and training time per block (middle)) at different sequence lengths and structure prediction performance with TM-score metric on five benchmarks (right).

processing longer protein sequences. For instance, as illustrated in Figure 1, attempting to process sequences of up to 800 in length would result in out-of-memory (OOM) issues. Consequently, the reduction of memory consumption in AlphaFold2 can be equated to the reduction of memory complexity in the biased attention mechanism, which is introduced by the communication between third-order tensors for MSA and pair-wise representation.

The diversity of protein sequences requires non-trivial attention to generality. However, previous linear attention mechanisms(Wang et al., 2020; Choromanski et al., 2021; Zhen et al., 2022) mainly use similarity decomposition to approximate Softmax($QK^T$) with different kernel functions, which reintroduce inductive bias and devise linear complexity at the expense of generality. Therefore, we turn to reconstructing the biased attention mechanism from a new perspective, a flow network perspective.

In this paper, we introduce a novel linear mechanism known as Bias-aware Flow Attention (BFA) within the framework of flow network theory. This mechanism efficiently transforms both third-order Multiple Sequence Alignment (MSA) and pair representations, reducing their complexity by an order of magnitude. Unlike previous linear attention methods that overlook bias matrices, BFA fully leverages biased information while linearizing the attention complexity. Empowered by BFA, our Liteformer can finally decrease memory and computational complexity from $\mathcal{O}(L^3 + sL^2)$ to $\mathcal{O}(L^2 + sL)$. In Figure 1, the left and middle panels depict the performance optimization with respect to both space and time across different sequence lengths.

We conducted extensive experiments on three monomeric and two multimeric protein datasets (as depicted in Figure 1 right panel). Our empirical results demonstrate the effectiveness of Liteformer in significantly reducing GPU memory consumption and training speed compared to Evoformer in AlphaFold2 (Jumper et al., 2021). Specifically, Liteformer substantially reduces memory usage by up to 44% and accelerates the training speed by 23% on a Nvidia A100 80G GPU, all while achieving competitive results in protein structure prediction against a strong baseline.

## 2 BACKGROUND

### 2.1 NOTATIONS

The input and output of Evoformer are two types of representation denoted as MSA representation $\mathbf{M} \in \mathbb{R}^{s \times L \times d_m}$ and pair representation $\mathbf{Z} \in \mathbb{R}^{L \times L \times d_z}$, where $s$ is the number of MSA sequence, $L$ is the length of protein sequence, $d_m, d_z$ are depth of dimension. Within each Evoformer block, $d_m, d_z$ are projected to $h \times d$ by canonical multi-head attention, where $h, d$ are head number and head dimension. The $i, j$ in the equations below are the subscription of the $L$ dimension.

### 2.2 EVOFORMER

This network has a two-tower architecture with axial self-attention for MSA representation $\mathbf{M}$ and triangular self-attention for pair representation $\mathbf{Z}$. Each Evoformer block has these representations as its input and output, and the outer product mean and the attention bias allow communication between representations, as illustrated in the top panel of Figure 2. In this section, we focus on a detailed analysis of the two multi-head biased attention modules within the Evoformer, which significantly impact memory utilization, namely MSA row-wise biased attention and pair triangular

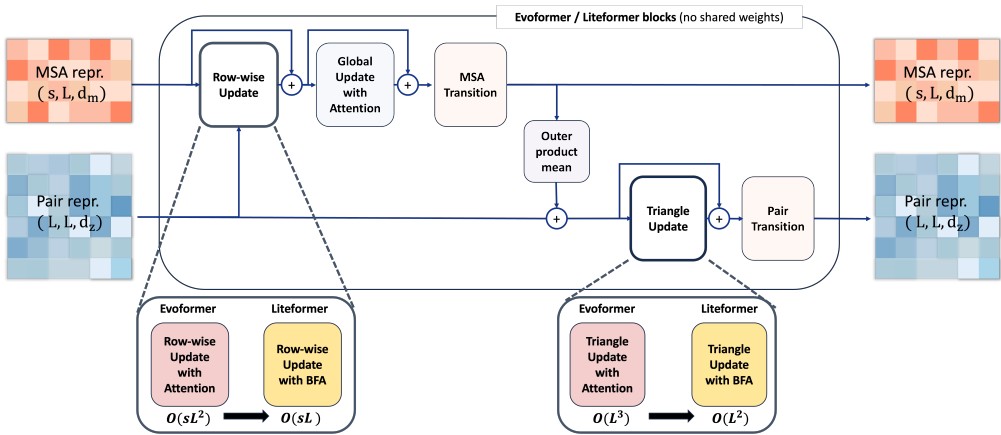

Figure 2: The structure comparison of Evoformer (in red) and our Liteformer (in yellow). In Evoformer blocks, pair representation and MSA representation are transformed by canonical attention and outer product mean and attention biasing allow the communication between these two representations. Compared to Evoformer, Liteformer transforms the MSA representation and the pair representation with Bias-aware Flow Attention (BFA). Similar to Evoformer blocks, transition and outer product mean are used for exchanging information between pair and MSA representation.

attention. The multi-head biased attention in the Evoformer is in the form of:

$$\text{Attention}(\mathbf{Q}, \mathbf{K}, \mathbf{V}, \mathbf{B}) = \text{Softmax}(\mathbf{Q}\mathbf{K}^T + \mathbf{B})\mathbf{V} \quad (1)$$

**Row-wise Biased Attention** For row-wise multi-head self-attention, MSA representation $\mathbf{M} \in \mathbb{R}^{s \times L \times d_m}$ is first projected into $\mathbf{Q}, \mathbf{K}, \mathbf{V} \in \mathbb{R}^{s \times h \times L \times d}$ and operates with pair representation projected as bias $\mathbf{B} \in \mathbb{R}^{1 \times h \times L \times L}$ in the form of Eq. 1, here $\mathbf{K}^T \in \mathbb{R}^{s \times h \times d \times L}$. The calculation of $\mathbf{Q}\mathbf{K}^T + \mathbf{B}$ causes $\mathcal{O}(shL^2d)$ complexity. Since $s, L$ is much larger than $h, d$, the total memory and computational complexity of the row-wise MSA representation update is $\mathcal{O}(sL^2)$.

**Triangular Update with Attention.** For triangular self-attention on the pair representation, it conducts row-wise and column-wise canonical multi-head biased self-attention. The update for each position $ij$ is modulated by both query-key similarity (i.e. $ij$ and $ik$) and the bias derived from the third position $jk$ to satisfy triangle inequality in geometry. Both attention mechanisms can be formalized as below. The pair representation $\mathbf{Z} \in \mathbb{R}^{L \times L \times d_z}$ is first projected into $\mathbf{Q}, \mathbf{K}, \mathbf{V} \in \mathbb{R}^{h \times L \times L \times d}$ and the triangle bias $\mathbf{B} \in \mathbb{R}^{h \times L \times L \times 1}$, the triangle attention result can be computed using Eq. 1, with $\mathbf{K}^T \in \mathbb{R}^{h \times L \times d \times L}$. The calculation of $(\mathbf{Q}\mathbf{K}^T + \mathbf{B}) \in \mathbb{R}^{h \times L \times L \times L}$ leads to $\mathcal{O}(L^3)$ memory and computational complexity. More details about row-wise and column-wise triangle attention can be seen in Appendix A.

After row-wise biased attention update, **global attention** calculates the similarity between the mean of queries along the $s$ dimension and keys to update the MSA column-wise information by canonical attention with $\mathcal{O}(sL)$ complexity. The resulting updated MSA representation is then transformed by **transition** with a fully connected layer, and an **outer product mean** block, which applies the outer product to each of two entries of MSA representation to generate an update for the corresponding pair representation entry. Please see more details in Appendix A.

It is notable that the bottleneck of Evoformer is the softmax function $\text{Softmax}(\mathbf{Q}\mathbf{K}^T + \mathbf{B})$ on third-order pair representation and MSA representation, which leads to $\mathcal{O}(L^3 + sL^2)$ complexity of the entire model. To tackle this obstacle, we present Liteformer as an alternative to the original Evoformer in AlphaFold2 to reduce memory and computational complexity from $\mathcal{O}(L^3 + sL^2)$ to $\mathcal{O}(L^2 + sL)$, as shown in Figure 2.

## 3 METHOD

### 3.1 LITEFORMER OVERVIEW

Given pair and MSA representations of an input protein sequence, our target is to optimize the computational complexity of Evoformer in AlphaFold2 (Jumper et al., 2021) for protein structure prediction. Inspired by flow network theory, the proposed Bias-aware Flow Attention (BFA) treats the biased attention mechanism as a flow network, where the flow originates from values nodes and merges into the biased attention result. The bias flow capacity and bias flow weights are determined by the query-key similarity as well as the bias matrix. Additionally, BFA conserves the flow capacity

to achieve a competition mechanism similar to Softmax. We define the conserved bias flow capacity and bias flow weights in Sec 3.2 and explain how Bias-aware Flow Attention achieves the biased attention effect in MSA and pair update with linear complexity in Sec 3.3.

## 3.2 CONSERVED BIAS FLOW

Ablation experiments on Evoformer (Jumper et al., 2021) demonstrate the substantial impact of bias on MSA row-wise update and pair-wise triangle update. However, previous linear attention mechanisms (Tay et al., 2022), whether Performer (Choromanski et al., 2021) and Cosformer (Zhen et al., 2022) using similarity decomposition kernels, or Sparse Transformer (Child et al., 2019) and Linformer (Wang et al., 2020) using attention sparsity, have never taken into account linearization with biased attention. Therefore, inspired by Flowformer (Wu et al., 2022a), we turn to analyze the attention calculation from a flow network perspective and seek to reconstruct the biased attention mechanism within this framework.

Previous work (Zhen et al., 2022) has identified two crucial characteristics of softmax that determine its performance in attention mechanism: a non-negative function on the attention matrix and a re-weighting mechanism to concentrate the distribution of attention connection. To achieve the effect of softmax, our Bias-aware Flow Attention features these two properties by non-negative flow capacity function $\phi(.)$ and competition mechanism among both source and sink nodes.

**Bias Flow Weights**. Since the Structure Module after the Evoformer only considers the primary sequence representation in MSA as sequence information for further prediction, we first design bias source nodes in this flow framework to expose the primary sequence with more pair-wise information for better structure prediction. The primary sequence and pair representation are first projected and reshaped to $\mathbf{Q}_0, \mathbf{K}_0, \mathbf{V}_0 \in \mathbb{R}^{h \times L \times d}$, and $\mathbf{B} \in \mathbb{R}^{h \times L \times L}$. Inspired by (Sun et al., 2023), we further introduce a relative position embedding (RoPE (Su et al., 2021)) on these projected tensors to enhance the model's extrapolation ability for a long sequence in inference time:

$$\mathbf{w_b} = \text{GroupNorm}((\mathbf{Q}_0 \odot \mathbf{\Theta})((\mathbf{K}_0 \odot \bar{\mathbf{\Theta}})^T + (\mathbf{B} \odot \mathbf{\Theta} \odot \bar{\mathbf{\Theta}}^T)) \tag{2}$$

where $\mathbf{\Theta} = [e^{l(i\theta)}]_{1 \times L \times 1}$ and its conjugation $\bar{\mathbf{\Theta}} = [e^{-l(i\theta)}]_{1 \times L \times 1}$, $\theta$ is a constant. $\odot$ denotes element-wise multiplication. This computation leads to $\mathcal{O}(hL^2d)$ memory and computational complexity. The biased primary sequence $\mathbf{w_b}\mathbf{V}_0$ are then projected and concatenated with the rest of the sequence projection to update queries $\mathbf{Q}'$, keys $\mathbf{K}'$ and biased source node $\mathbf{V}'$:

$$\mathbf{Q}' = [(\mathbf{w_b}\mathbf{V}_0)\mathbf{W_Q}; \mathbf{Q}_{1:}]; \quad \mathbf{K}' = [(\mathbf{w_b}\mathbf{V}_0)\mathbf{W_K}; \mathbf{K}_{1:}]; \quad \mathbf{V}' = [(\mathbf{w_b}\mathbf{V}_0)\mathbf{W_V}; \mathbf{V}_{1:}]$$

where $\mathbf{W_Q}, \mathbf{W_K}, \mathbf{W_V} \in \mathbb{R}^{d \times d}$, $\mathbf{Q}', \mathbf{K}', \mathbf{V}' \in \mathbb{R}^{s \times h \times L \times d}$, and $[;]$ denotes tensor concatenation.

**Similarity Score.** In general, the similarity between queries $\mathbf{Q}'$ and keys $\mathbf{K}'$ with bias $\mathbf{B}$ is a function of $\mathbf{Q}'\mathbf{K}'^T + \mathbf{B}$, in terms of $\mathcal{F}(\mathbf{Q}'\mathbf{K}'^T + \mathbf{B})$. Bias-aware flow attention defines this function with a non-linear and non-negative function $\phi(\cdot) = \text{Sigmoid}(\cdot)$. The similarity score is as follows:

$$SM(\mathbf{Q}', \mathbf{K}', \mathbf{B}) = \mathcal{F}(\mathbf{Q}'\mathbf{K}'^T + \mathbf{B}) = \phi(\mathbf{Q}')\phi(\mathbf{K}')^T + \phi(\mathbf{B}) \tag{3}$$

**Bias Flow Capacity.** As we mentioned above, there are two types of information flow, incoming flow from the aggregation of values and outgoing flow to contribute to the results. $\mathbf{I}_i$, incoming flow to sink $i$ from source nodes, and $\mathbf{O}_j$, the outgoing flow from source $j$ to sink nodes, can be defined as follows:

$$\mathbf{I}_i = \phi(\mathbf{Q}'_i) \sum_{j=1}^{L} \phi(\mathbf{K}'_j)^T + \sum_{j=1}^{L} \phi(\mathbf{B}_{ij}), \quad \mathbf{O}_j = \phi(\mathbf{K}'_j) \sum_{i=1}^{L} \phi(\mathbf{Q}'_i)^T + \sum_{i=1}^{L} \phi(\mathbf{B}_{ij}) \tag{4}$$

where $\phi(\mathbf{Q}'_i), \phi(\mathbf{K}'_j) \in \mathbb{R}^{s \times h \times d}$, $\phi(\mathbf{B}_{ij}) \in \mathbb{R}^h$, $\mathbf{I}_i, \mathbf{O}_j \in \mathbb{R}^{s \times h}$. Since we first sum along $L$ dimension, the computation of $\mathbf{I}, \mathbf{O}$ has $\mathcal{O}(shLd)$ memory and computational complexity.

**Conserved Bias Flow Capacity.** Softmax function is originally proposed as a "winner-take-all" picking maximum operation (Bridle, 1989), enforcing higher attention only to the essential tokens. To introduce this "winner-take-all" mechanism in a flow network, we aim to conserve the capacity of incoming and outgoing flow, resulting in competition among both the source and sink nodes to

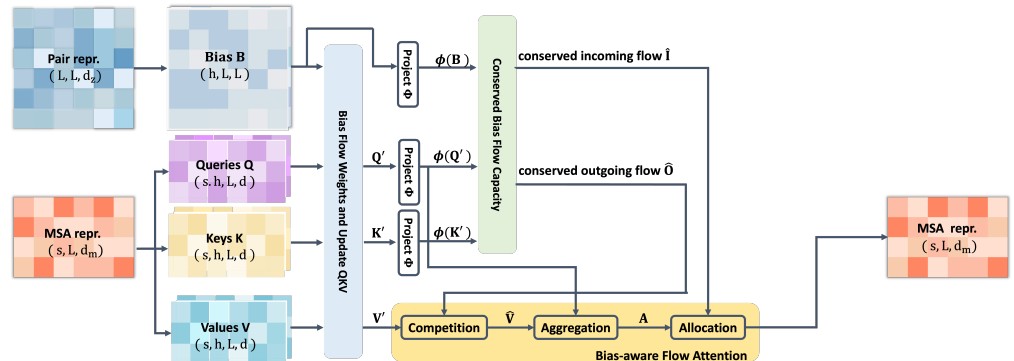

Figure 3: Pipeline of Bias-aware Flow Attention (BFA) for MSA row-wise update. MSA representation and pair representation are respectively multi-head projected into queries $\mathbf{Q}$, keys $\mathbf{K}$, values $\mathbf{V}$ and bias $\mathbf{B}$ matrix. The primary sequence projections $\mathbf{Q}_0, \mathbf{K}_0, \mathbf{V}_0$ first incorporate with bias $\mathbf{B}$ to calculate bias flow weights for their update, and then concatenate with the remaining projections. A non-linear function $\phi(\cdot) = \text{Sigmoid}(\cdot)$ transforms the concatenated queries $\mathbf{Q}'$ and keys $\mathbf{K}'$ to generate conserved incoming flow $\hat{\mathbf{I}}$ and conserved outgoing flow $\hat{\mathbf{O}}$. This flow information calculates the final result from concatenated values $\mathbf{V}'$ through competition, information aggregation, and allocation.

concentrate the distribution of the attention mechanism. We set the flow capacity as default 1 and get the conserved incoming and outgoing flow defined as:

$$\widehat{\mathbf{I}}_i = \phi(\mathbf{Q}'_i) \sum_{j=1}^{L} \frac{\phi(\mathbf{K}'_j)^T}{\mathbf{O}_j} + \sum_{j=1}^{L} \frac{\phi(\mathbf{B}_{ij})}{\mathbf{O}_j}, \quad \widehat{\mathbf{O}}_j = \phi(\mathbf{K}'_j) \sum_{i=1}^{L} \frac{\phi(\mathbf{Q}'_i)^T}{\mathbf{I}_i} + \sum_{i=1}^{L} \frac{\phi(\mathbf{B}_{ij})}{\mathbf{I}_i} \quad (5)$$

This normalization also leads to a $\mathcal{O}(shLd)$ memory and computational complexity. Please see more details in Appendix B.

### 3.3 BIASED-AWARE FLOW ATTENTION

Our bias-aware flow attention is based on multi-head attention working for third-order tensors with a given bias and its complexity can achieve $\mathcal{O}(L)$. After defining the conserved information flow and bias source nodes $\mathbf{V}'$, our bias-aware flow attention performs the effect of attention in three steps: competition, information aggregation, and allocation. For MSA row-wise update, the MSA representation can be first projected to $\mathbf{Q}', \mathbf{K}', \mathbf{V} \in \mathbb{R}^{s \times h \times L \times d}$, where $h, d$ are the head number and head dimension, and the pair representation is also projected and reshaped to $\mathbf{B} \in \mathbb{R}^{h \times L \times L}$,.

**Competition.** For the biased source nodes ($\mathbf{V}'$), the competition mechanism conserves outgoing flow capacity to implement a "winner-takes-all" effect, determining the gating weights for each biased source node:

$$\widehat{\mathbf{V}} = \text{Softmax}_L(\widehat{\mathbf{O}}) \odot \mathbf{V}' \quad (6)$$

where $\widehat{\mathbf{O}} \in \mathbb{R}^{s \times h \times L}, \hat{\mathbf{V}} \in \mathbb{R}^{s \times h \times L \times d}$ and $\odot$ denotes broadcast element-wise multiplication, the memory and computational complexity of which is $\mathcal{O}(shLd)$.

**Aggregation.** Information aggregation is implemented with competitive source nodes. Similar to Row-wise attention (Eq. 1), we compute the distribution of incoming flow with the help of competitive source nodes $\hat{\mathbf{V}}$ and similarity score $\phi(\mathbf{Q}')\phi(\mathbf{K}')^T$:

$$\mathbf{A} = \frac{(\phi(\mathbf{Q}')\phi(\mathbf{K}')^T)\widehat{\mathbf{V}}}{\phi(\mathbf{Q}') \sum_j \phi(\mathbf{K}'_j)^T} = \frac{\phi(\mathbf{Q}')}{\phi(\mathbf{Q}') \sum_j \phi(\mathbf{K}'_j)^T}(\phi(\mathbf{K}')^T\widehat{\mathbf{V}}) \quad (7)$$

By first computing the matrix product of $\phi(\mathbf{K}')^T\widehat{\mathbf{V}} \in \mathbb{R}^{s \times h \times d \times d}$, the memory and computational complexity of $\mathbf{A}$ reduced from $\mathcal{O}(shL^2d)$ to $\mathcal{O}(shLd^2)$.

**Allocation.** We finally introduce competition among sink nodes ($RowAttn$) with the use of conserved incoming flow as a gate function and the memory and computational complexity of this operation is still $\mathcal{O}(shLd)$:

$$RowAttn = \text{Sigmoid}(\widehat{\mathbf{I}}) \odot \mathbf{A} \quad (8)$$

where $\odot$ denotes broadcast element-wise multiplication. The memory and computational complexity of the entire Bias-aware Flow Attention finally achieves $\mathcal{O}(shLd^2 + shLd + hL^2d) \approx \mathcal{O}(sL)$, when $s, L$ are at the same magnitude and much larger than $h, d$

For pair representation update, both the row-wise and column-wise triangle attention can be linearized as below. The pair representation $\mathbf{Z} \in \mathbb{R}^{L \times L \times c_z}$ is projected and reshape into $\mathbf{Q}, \mathbf{K}, \mathbf{V} \in \mathbb{R}^{h \times L \times L \times d}$ and triangle bias $\mathbf{B}_{tri} \in \mathbb{R}^{h \times L \times L \times 1}$. The $(i, j, k)$ th biased similarity score is in the form of:

$$SM(\mathbf{Q}_{ij}, \mathbf{K}_{ik}, (\mathbf{B}_{tri})_{jk}) = \mathcal{F}(\sum_t^d q_t \odot k_t + \mathbf{1}_h \odot b) = \mathcal{F}(\sum_t^{d+1} q'_t \odot k'_t) \qquad (9)$$

where $q_t, k_t, \mathbf{1}_h \in \mathbb{R}^{h \times 1}$, $q'_{(d+1)} = \mathbf{1}_h, k'_{(d+1)} = b, \odot$ denotes element-wise multiplication. Thus, Eq. 3 can be easily reformulated to an unbiased similarity score by tensor concatenation.

$$SM(\mathbf{Q}', \mathbf{K}', \mathbf{B}_{tri}) = \mathcal{F}(\mathbf{Q}'\mathbf{K}'^T + \mathbf{B}_{tri}) = \phi(\mathbf{Q}')\phi(\mathbf{K}')^T \qquad (10)$$

where $\mathbf{Q}' = [\mathbf{Q}; \mathbf{1}_L], \mathbf{K}' = [\mathbf{K}; \mathbf{B}_{tri}], [;]$ denotes matrix concatenation and $\mathbf{1}_L \in \mathbb{R}^{h \times L \times L \times 1}$, $\mathbf{Q}', \mathbf{K}' \in \mathbb{R}^{h \times L \times L \times (d+1)}$. Since triangle bias only has a single $L$ dimension on the last two dimensions, the concatenated queries and keys are capable of computing unbiased flow capacity in Eq. 4 and introduce the competitive effect similar to softmax by controlling total flow capacity in Eq. 5. Following the same procedure in Eq. 6, 7 and 8, pair representation triangular update decreases its memory and computational complexity from $\mathcal{O}(L^3)$ to $\mathcal{O}(L^2)$. The pipeline of triangular update with BFA as well as the contrast between BFA and canonical attention pipeline can be seen in Appendix C.

## 4 EXPERIMENTS

### 4.1 EXPERIMENTAL SETUP

**Datasets.** We compare the performance of our Liteformer and baseline Evoformer (AlphaFold2) on three monomeric datasets: CASP14 (Moult et al., 2020), CASP15 (Moult et al., 2022),CAMEO (X. et al., 2021), and two complex datasets:VH-VL, DB5.5 (Vreven et al., 2015).

- **CASP14** & **CASP15.** The 14th & 15th Critical Assessment of protein Structure Prediction, which contains 50 and 56 single protein targets with sequence length less than 1000.
- **CAMEO.** The Continuous Automated Model Evaluation, which includes 194 single protein targets with sequence length ranging from 60 to 800.
- **VH-VL.** This complex protein dataset includes approximately 4.7k antibodies from PDB. We randomly selected 159 samples as the test set. The sequence length of each sample is less than 450.
- **DB5.5.** It is a multi-domain protein dataset with experimentally determined structures covering a range of difficulty levels for structure prediction. We select 12 complex targets following the experimental setting of EquiDock (Ganea et al., 2021).

**Evaluation Metrics.** We gauge the computational cost of training each Evoformer/Liteformer block through measures of memory usage (**Memory per Block**) and time consumption (**Cost per Block**). In assessing the structural prediction performance comprehensively, we employ metrics including **TM-score**, **RMSD**, **GDT-TS**, and **GDT-HA** for structural prediction accuracy, along with **DockQ** for evaluating docking performance. More details about each evaluation metric can be seen in Appendix D.2.

**Implementation.** We implement our Liteformer based on OpenFold (Ahdritz et al., 2022) Framework. We trained 10,000 randomly selected general protein data from PDB for 5 days using 8 × DGX-A100 80G GPUs and inference on two multimeric datasets: VH-VL and DB5.5. We further trained a combined model of Liteformer and ESM-2 150M (Lin et al., 2022b) with 400,000 general single protein data for 5 days and inference on single protein targets selected from three datasets: CASP14, CASP15 and CAMEO. Corresponding to Evoformer in AlphaFold2, our Liteformer contains 48 blocks. We set $L$ in the training stage as 256 for the single protein structure prediction task and 384 for the complex protein structure prediction task. We set MSA sequence number $s$ as 1024,

Table 1: Memory and computational cost comparison for each module component.

| Module | Mem per block (GB) | | Cost per block (ms) | |
|---|---|---|---|---|
| | Evoformer | Liteformer | Evoformer | Liteformer |
| MSA row-wise update | 25.318 | **14.931** | 197.657 | **140.435** |
| Pair triangle update | 3.658 | **1.560** | 41.253 | **39.417** |

Table 2: Quantitative results comparison on CASP and CAMEO benchmarks.

| Dataset | Model w/ ESM-2 | TM-score ↑ | RMSD ↓ | GDT-TS ↑ | GDT-HA ↑ |
|---|---|---|---|---|---|
| CASP14 | Evoformer | **0.4971** | 10.086 | **0.2535** | **0.1347** |
| | Liteformer | 0.4857 | **9.663** | 0.2383 | 0.1220 |
| CASP15 | Evoformer | 0.5356 | 18.663 | 0.2433 | 0.1269 |
| | Liteformer | **0.5517** | **17.594** | **0.2536** | **0.1377** |
| CAMEO | Evoformer | 0.7112 | 5.962 | 0.4477 | 0.2770 |
| | Liteformer | **0.7177** | **5.558** | **0.4627** | **0.2901** |

head number $h$ as 8, and head dimension $d$ as 32. Our baseline is ESM-2 150M with Evoformer for a single protein structure prediction task, and AlphaFold-Multimer with Evoformer for a complex protein structure prediction task. We also conduct AlphaFold2 as baseline for single structure prediction, the results can be seen in Appendix E. We finally compare the computational cost, such as memory and time consumption, and accuracy of predicted 3D structure between our Liteformer and Evoformer. The training loss can be seen in Appendix D.1. Since Flash Attention (Dao et al., 2022) lacks the capability to expedite computation with a predetermined bias, we opt not to compare the efficiency of our model with it.

## 4.2 EXPERIMENTAL RESULTS

**Computational Cost.** Table 1 compares the memory and time consumption of different modules, including MSA row-wise update and Pair triangle update, in Evoformer and Liteformer block. The first two columns show that our Liteformer substantially diminishes memory by 43.67% for MSA update and 57.35% for pair representation update. Meanwhile, the training time of our Liteformer can be decreased up to 28.95%. Figure 1 (Left, Middle) shows the memory and time consumption of Evoformer and Liteformer at different sequence lengths. As the sequence length increases, the effect of our model's reduction in memory usage becomes more clear. In sequence length of 600, our Liteformer significantly reduces memory consumption by 53.33% for the entire model. Our Liteformer even handles sequence lengths up to 800, while the Evoformer experiences out-of-memory (OOM). Additionally, the speed of our Liteformer increases as the sequence length becomes longer. The time consumption of Liteformer can be reduced by 21.23%.

**Structure Prediction on Multiple Benchmarks.** Table 2 & 3 shows the quantitative comparisons with Evoformer on structure predictions. Table 2 evaluates three single protein structure datasets, CASP14, CASP15 and CAMEO, and qualifies the performance with several metrics, such as TM-score and RMSD for structure accuracy. It is evident that the accuracy of the structure predicted by Liteformer with ESM-2 150M is comparable to Evoformer with ESM-2 150M on CASP14 and even surpasses the baseline model on CASP15 and CAMEO. The Appendix E provides a comparative curve of our Liteformer's performance and that of the baseline across these three datasets, tracking their progress as training steps grow. Table 3 assesses the performance of our Liteformer against AlphaFold-Multimer on two complex protein sequence datasets, VH-VL and DB5.5 using various metrics, such as TM-score for structure accuracy and DockQ for binding site accuracy. We can observe that the accuracy of the structure predicted by Liteformer is comparable to AlphaFold-Multimer on VH-VL and DB5.5 benchmarks.

## 4.3 ABLATION STUDIES

In this section, we performed ablation studies to demonstrate the benefit of introducing Bias-aware attention modules. We also conducted extensive experiments to explore the effect of bias flow weight and bias flow weight in Bias-aware attention, and the training sequence length in Liteformer.

**Bias-aware Flow Attention.** Table 4 ablates the effect of Bias-aware Flow Attention on MSA row-wise update and pair triangle update. Here we set the sequence length $L$ as 256 and MSA number $s$ as 1024. We compare three models, AlphaFold2, Liteformer, and Liteformer without BFA in

Table 3: Quantitative results comparison on VH-VL and DB5.5 benchmarks.

| Dataset | Model w/ AF2-Multimer | TM-score ↑ | DockQ ↑ | RMSD ↓ |
|---------|----------------------|------------|---------|--------|
| VH-VL | Evoformer | 0.9662 | 0.7168 | 1.325 |
|       | Liteformer | **0.9668** | **0.7514** | **1.302** |
| DB5.5 | Evoformer | 0.7314 | 0.6404 | 7.461 |
|       | Liteformer | 0.7258 | **0.6927** | **7.367** |

Table 4: Ablation studies on the effect of Bias-aware Flow Attention (BFA) on MSA update and Pair update in terms of computation and prediction performance on VH-VL benchmark.

| MSA | Pair | Mem per block (G) | Cost per block(ms) | TM-score ↑ | DockQ ↑ | RMSD ↓ |
|-----|------|-------------------|--------------------|-----------|---------|--------|
| ✗ | ✗ | 28.976 | 436.566 | 0.9640 | 0.7352 | 1.295 |
| ✗ | ✓ | 26.878 | 434.761 | 0.9647 | 0.7488 | 1.264 |
| ✓ | ✓ | 16.323 | 336.522 | 0.9638 | 0.7376 | 1.278 |

MSA row-wise update, and evaluate the prediction performance on the VH-VL benchmark, and the memory consumption and time cost per Liteformer block.

Compared to the performance of AlphaFold2, it is notable that Liteformer benefits more from BFA in pair updates. BFA in pair triangular update (on the second row) increases TM-score from 0.9640 to 0.9647 and DockQ from 0.7352 to 0.7488. Although canonical biased attention for MSA row-wise update allows Liteformer to sacrifice less information for better results, it requires larger memory usage and time consumption. Thus, there is a trade-off between performance and memory usage. In our model, we set Bias-aware Flow Attention on both MSA and Pair update to balance this tradeoff. Notably, the proposed Liteformer (on the last row) can achieve a significant reduction in computational consumption while maintaining comparable performance to the vanilla baseline.

**Bias Flow Weight & Flow Capacity.** To validate the effectiveness of introducing bias in our Bias-aware Flow Attention (BFA) module, in Table 5, we compare three models: BFA, BFA without bias flow weight, and BFA without bias flow capacity in terms of memory and time usage as well as performance on VH-VL datasets. Here we set the Liteformer block number to 8 and sequence length to 256. For BFA without bias flow capacity (on the first row), we do not incorporate bias terms in calculating conserved flow capacity, which results in a reduction of memory consumption by more than half of BFA. The reason for memory reduction is that the intermediate variables related to bias are no longer required to be retained for gradient backward propagation. Compared to the performance of BFA (in the last row), it is obvious that the performance advantages of BFA are more attributed to bias flow capacity. This is intuitively plausible, as the computation of bias flow capacity allows bias (the pair-wise information) to incorporate with all sequences in MSA. For BFA without bias flow weight, we only introduce bias by bias flow capacity. Our Bias-aware Flow Attention takes advantage of both bias flow weight and bias flow capacity, and reaches a better performance than baseline Evoformer in AlphaFold2 with less time and memory usage.

**Impact of sequence length $L$ in Liteformer.** In Figure 4, we discuss the role of hyper-parameter sequence length $L$ in Liteformer for improving structure prediction. Here we compare the performance of Liteformer with 8 Liteformer blocks under sequence lengths of 256, 350 and 400 on the VH-VL benchmark. To compare clearly, we only capture the images from the final stage where models are approaching convergence. We can observe that both the accuracy metric (TM-score) and docking metric (DockQ) increase as the value of $L$ increases, which aligns with our intuition that the longer the sequence, the more knowledge the model can learn. Compared to a sequence length of

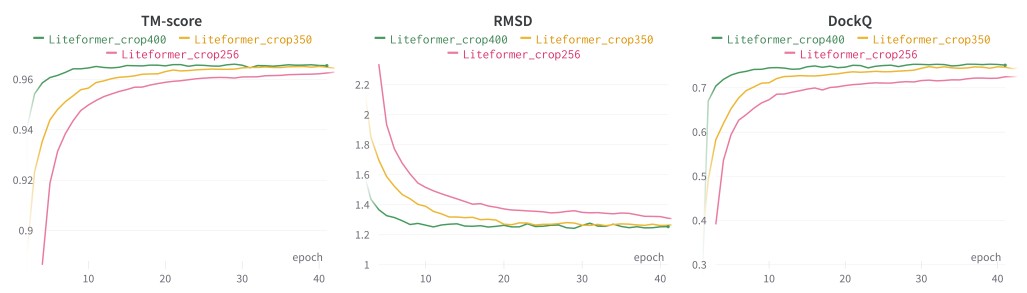

Figure 4: Performance of Liteformer with sequence length 256, 350 and 400 on VH-VL dataset.

Table 5: Ablation studies on bias flow capacity and bias flow weights of Bias-aware Flow Attention (BFA) in terms of computation and prediction performance on the VH-VL benchmark.

| Model | Mem per block (G) | Cost per block(ms) | TM-score ↑ | DockQ ↑ | RMSD ↓ |
|---|---|---|---|---|---|
| BFA w/o bias flow capacity | 5.366 | 325.253 | 0.9606 | 0.7201 | 1.341 |
| BFA w/o bias flow weights | 16.491 | 371.892 | 0.9608 | 0.7282 | 1.322 |
| **BFA** | **16.323** | **336.522** | **0.9638** | **0.7376** | **1.278** |

256, Liteformer with 400 sequence length has seen an increase in DockQ from 0.722 to 0.753, TM score from 0.9626 to 0.9655, and the RMSD metric decreases from 1.311 to 1.255. Additionally, the speed of convergence accelerates as the sequence length increases.

## 5 RELATED WORK

**Protein Structure Prediction.** Protein structure prediction aims to predict protein 3D structure using a primary sequence as input, which has been addressed in many previous works (Heo & Feig, 2018; Huang, 2017; Senior et al., 2020; Wang et al., 2017). Typically, researchers in DeepMind proposed AlphaFold2 (Jumper et al., 2021), a transformer-based model to achieve a seminal work. The pipeline of AlphaFold2 can be split into three parts: data preparation, Evoformer, and structure module. In the first part of the pipeline, based on input sequence, the model searches multiple sequence alignment (MSA) and determined structure template (in terms of pair-wise representations), which contain valuable evolutionary and structural information respectively to assist further structure modeling. Since searching for MSA is time costly and consumes large memory, some subsequent models, such as Omegafold (Wu et al., 2022b), ESMfold (Lin et al., 2022b) and HelixFold (Fang et al., 2022), attempt to substitute MSA searching with pre-trained protein language model (PLM), to get sequence representation. Compared to these works, our model focuses on reducing memory consumption in the representation learning stage and is also fairly compatible with pre-trained language models, such as ESM-2.

**Efficient Linear Transformer.** To break through the computational limitation caused by the quadratic complexity of pair-wise relation modeling, various efficient linear transformers (Ho et al., 2019; Iz et al., 2020; Ma et al., 2021; Vyas et al., 2020; Zhang et al., 2021) have been explored. One category of methods (Child et al., 2019; Wang et al., 2020; Kitaev et al., 2020; Zaheer et al., 2020; Lepikhin et al., 2020; Ding et al., 2023) attempts to reduce the model's captured relations by utilizing sparsity. By sparsely activating a subset of parameters, these models can achieve lower complexity at the expense of information loss, leading to the tradeoff between performance and efficiency. More recently, RetNet (Sun et al., 2023) supported a hybrid form of recurrent and parallel computation manner, which allows queries to attend all previous keys with linear complexity. Another mainstream of efficient transformers (Katharopoulos et al., 2020; Choromanski et al., 2021; Zhen et al., 2022) is similarity decomposition methods, which try to substitute or approximate the softmax similarity by designing a non-linear kernel function. Flowformer (Wu et al., 2022a) utilized a simple non-negative projection function but brought flow conservation into the design to achieve competition among tokens. However, the aforementioned methods mainly focus on the approximation of the unbiased similarity metric, and none of them attempt to linearize a more general similarity score with a given bias. Unlike previous works, our model manages to linearize biased attention for third-order tensors by similarity decomposition.

## 6 CONCLUSION

We present Liteformer, a novel and lightweight architecture to reduce the complexity of Evoformer from $\mathcal{O}(L^3 + sL^2)$ to $\mathcal{O}(L^2 + sL)$. The proposed framework leverages a Bias-aware Flow Attention mechanism to linearize biased attention with $\mathcal{O}(L)$ complexity. Our Liteformer decreases memory consumption by significant margins while achieving competitive prediction results compared to the strong baseline Evoformer in AlphaFold2. In a broader context, various domains, including scoring models (Liu et al., 2023), protein function prediction models (Hu et al., 2022) and docking prediction (Luo et al., 2023), have integrated the Evoformer module. The improvement of our Liteformer holds the potential to further advancement in these areas.

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

APPENDIX

In this appendix, we introduce sub-modules of Evoformer in Sec A and provide detailed Bias-aware Flow Attention, including conserved bias flow capacity in Sec B and triangular update pipeline in Sec C. We also introduce experimental setup in Sec D and more experimental results are in Sec E

## A  EVOFORMER SUBMODULES

**Global Attention.**  The MSA representation is first projected and reshaped into $\mathbf{Q}, \mathbf{K}, \mathbf{V} \in \mathbb{R}^{h \times L \times s \times d}$. Global Attention uses the mean of queries along $s$ dimension to compute attention as below:

$$GlobalAttn(\mathbf{Q}, \mathbf{K}, \mathbf{V}) = \text{Softmax}(\text{mean}_s(\mathbf{Q})\mathbf{K}^T)\mathbf{V} \tag{11}$$

where $\text{mean}_s(\mathbf{Q}) \in \mathbb{R}^{h \times L \times 1 \times d} \mathbf{K}^T \in \mathbb{R}^{h \times L \times d \times s}$. The calculation of softmax leads to $\mathcal{O}(hLsd) \approx \mathcal{O}(Ls)$ when $s, L$ are much larger than $h, d$.

**Triangular Update with Attention**  By considering residues in protein sequences as graph nodes, the pair-wise information can be treated as the edge information between residue pairs. The row-wise triangular attention updates the edges $ij$ with values from all edges that share the same starting node $i$.(i.e. all edges $ik$). The contribution of each edge $ik$ is determined by both query-key similarity and the bias information $b_{jk}$ derived from edge $ij$. The pair representation is projected into $\mathbf{Q}, \mathbf{K} \in \mathbb{R}^{h \times L \times L \times d}$ and the triangle bias $\mathbf{B} \in \mathbb{R}^{h \times L \times L \times 1}$. The similarity score of the row-wise triangular update is:

$$RowSM_{ij} = \text{Softmax}(\mathbf{Q}_{ij}^T \mathbf{K}_{ik} + \mathbf{B}_{jk}) \tag{12}$$

where $\mathbf{Q}_{ij}^T \in \mathbb{R}^{h \times 1 \times d}, \mathbf{K}_{ik} \in \mathbb{R}^{h \times d \times 1}$.

Similarly, column-wise triangular attention aggregates values from all edges that share the same ending nodes $j$(i.e.$kj$) to update edge $ij$ , the weight of each edge can be modulated by query-key similarity as well as the bias information from the third edges $ki$.

$$ColumnSM_{ij} = \text{Softmax}(\mathbf{Q}_{ij}^T \mathbf{K}_{kj} + \mathbf{b}_{ki}) \tag{13}$$

**MSA Transition** & **Pair Transition.**  After row-wise and global attention, MSA representation $\mathbf{M}$ is transformed by 2-layer MLP blocks in MSA Transition.

$$\mathbf{M} = (\text{RELU}(\mathbf{M}\mathbf{W}_{MLP1}))\mathbf{W}_{MLP2} \tag{14}$$

where $\mathbf{W}_{MLP1} \in \mathbb{R}^{d_m \times 4d_m}, \mathbf{W}_{MLP2} \in \mathbb{R}^{4d_m \times d_m}$. After row-wise and column-wise triangular attention update, we also transform pair representation $\mathbf{Z}$ by 2-layer MLP blocks in Pair Transition.

$$\mathbf{Z} = (\text{RELU}(\mathbf{Z}\mathbf{W}_{MLP1}))\mathbf{W}_{MLP2} \tag{15}$$

where $\mathbf{W}_{MLP1} \in \mathbb{R}^{d_z \times 4d_z}, \mathbf{W}_{MLP2} \in \mathbb{R}^{4d_z \times d_z}$.

**Outer Product Mean.**  The outer product mean block transforms the MSA representation into an update for the pair representation. For updating entry $(i, j)$ of pair representation, the outer products of vectors from two columns $i$ and $j$ are averaged over the sequences. Then the product is flattened and projected to dimension $d_z$ for the update.

$$\widehat{\mathbf{Z}}_{ij} = \text{Flatten}(mean_s(\mathbf{M}_i \otimes \mathbf{M}_j)\mathbf{W}_z \tag{16}$$

where $\mathbf{M}_i, \mathbf{M}_j \in R^{s \times d_m}, \mathbf{M}_i \otimes \mathbf{M}_j \in R^{s \times d_m \times d_m}, W_z \in R^{d_m^2 \times d_z}$, $\otimes$ denotes the outer product. The pair representation is updated with MSA information by outer product mean:

$$\mathbf{Z}_{ij} = \mathbf{Z}_{ij} + \widehat{\mathbf{Z}}_{ij} \tag{17}$$

## B  DETAILED CONSERVED BIAS FLOW CAPACITY

Regarding conserved bias flow capacity, We introduce a competition mechanism into the flow network by limiting the flow capacity. In the simple scenario, we set flow capacity as default 1 for each source and sink node:

$$\frac{\phi(\mathbf{Q}_i')\sum_{j=1}^{L}\phi(\mathbf{K}_j')^T + \sum_{j=1}^{L}\phi(\mathbf{B}_{ij})}{\mathbf{I}_i} = 1, \quad \frac{\phi(\mathbf{K}_j')\sum_{i=1}^{L}\phi(\mathbf{Q}_i')^T + \sum_{i=1}^{L}\phi(\mathbf{B}_{ij})}{\mathbf{O}_j} = 1 \tag{18}$$

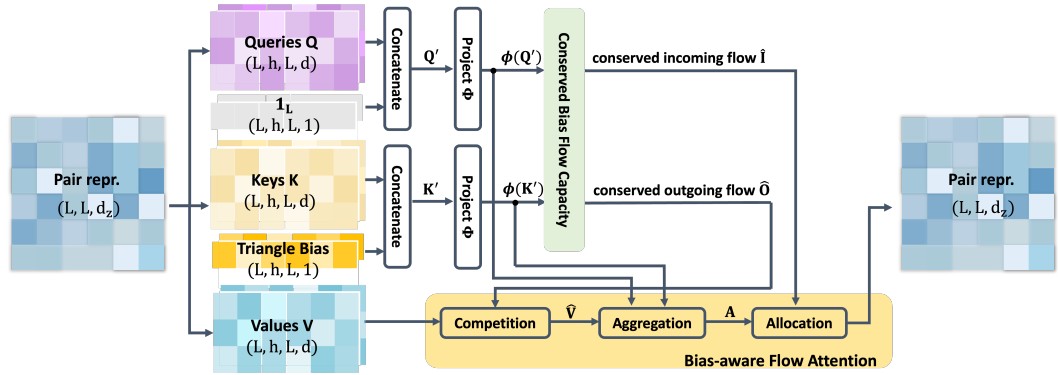

Figure 5: Pair representation row-wise and column-wise triangle update through Bias-aware Flow Attention.

By treating $\frac{\phi(\mathbf{Q}'_i)}{\mathbf{I}_i}, \frac{\phi(\mathbf{K}'_j)}{\mathbf{O}_j}$ as normalization operations for queries-keys similarity and $\frac{\phi(\mathbf{B}_{ij})}{\mathbf{I}_i}, \frac{\phi(\mathbf{B}_{ij})}{\mathbf{O}_j}$ as normalization operations for bias matrix, and applying the normalization to Eq. 4, we get the conserved incoming and outgoing flow:

$$\widehat{\mathbf{I}}_i = \phi(\mathbf{Q}'_i) \sum_{j=1}^{L} \frac{\phi(\mathbf{K}'_j)^T}{\mathbf{O}_j} + \sum_{j=1}^{L} \frac{\phi(\mathbf{B}_{ij})}{\mathbf{O}_j}, \quad \widehat{\mathbf{O}}_j = \phi(\mathbf{K}'_j) \sum_{i=1}^{L} \frac{\phi(\mathbf{Q}'_i)^T}{\mathbf{I}_i} + \sum_{i=1}^{L} \frac{\phi(\mathbf{B}_{ij})}{\mathbf{I}_i} \qquad (19)$$

## C  TRIANGULAR UPDATE WITH BIAS-AWARE FLOW ATTENTION

As shown in Figure 5, pair representation is first projected and reshaped into $\mathbf{Q}, \mathbf{K}, \mathbf{V} \in \mathbb{R}^{h \times L \times L \times d}$ and triangle bias $\mathbf{B} \in \mathbb{R}^{h \times L \times L \times 1}$. The queries and keys matrix first concatenates with $\mathbf{1}$ and triangle bias. Then based on the unbiased version of Eq **??**, we use the concatenated queries $\mathbf{Q}'$ and keys $\mathbf{K}'$ to calculate the conserved incoming and outgoing flow.

$$\mathbf{I}_i = \phi(\mathbf{Q}'_i) \sum_{j=1}^{L} \phi(\mathbf{K}'_j)^T, \quad \mathbf{O}_j = \phi(\mathbf{K}'_j) \sum_{i=1}^{L} \phi(\mathbf{Q}'_i)^T \qquad (20)$$

$$\widehat{\mathbf{I}}_i = \phi(\mathbf{Q}'_i) \sum_{j=1}^{L} \frac{\phi(\mathbf{K}'_j)^T}{\mathbf{O}_j}, \quad \widehat{\mathbf{O}}_j = \phi(\mathbf{K}'_j) \sum_{i=1}^{L} \frac{\phi(\mathbf{Q}'_i)^T}{\mathbf{I}_i} \qquad (21)$$

Following the same procedure of Bias-aware Flow Attention on MSA, the triangular update can be implemented within three steps: competition, aggregation and allocation.

$$\widehat{\mathbf{V}} = \text{Softmax}_L(\widehat{\mathbf{O}}) \odot \mathbf{V}$$

$$\mathbf{A} = \frac{(\phi(\mathbf{Q}')\phi(\mathbf{K}')^T)\widehat{\mathbf{V}}}{\phi(\mathbf{Q}') \sum_j \phi(\mathbf{K}'_j)^T} = \frac{\phi(\mathbf{Q}')}{\mathbf{I}} (\phi(\mathbf{K}')^T \widehat{\mathbf{V}}) \qquad (22)$$

$$TriangleAttn = \text{Sigmoid}(\widehat{\mathbf{I}}) \odot \mathbf{A}$$

## D  EXPERIMENTAL SETUP

### D.1  TRAINING LOSS

During training, we apply the same objective function as AlphaFold2 (Jumper et al., 2021). The primary reason to use various loss function terms is to attach an individual loss to each major sub-component of the model (including both the pair and MSA final embeddings) as a guide during the training.

$$\mathcal{L} = 0.5\mathcal{L}_{FAPE} + 0.5\mathcal{L}_{aux} + 0.3\mathcal{L}_{dist} + 2\mathcal{L}_{msa} + 0.01\mathcal{L}_{conf} \qquad (23)$$

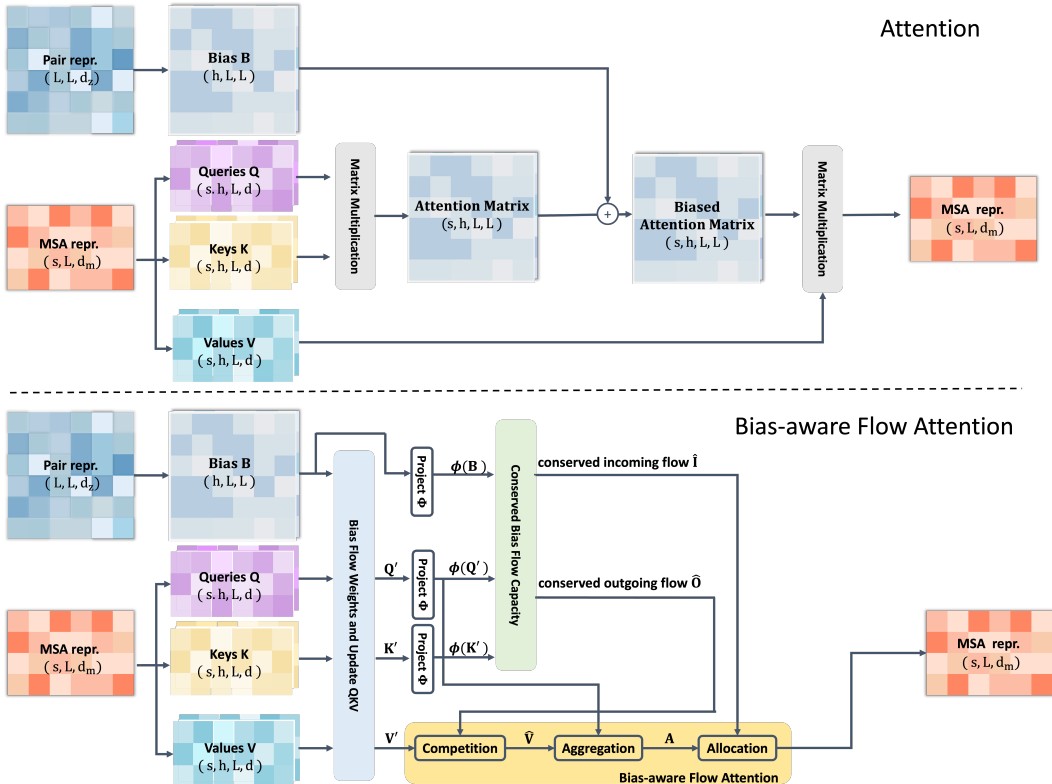

Figure 6: The contrast between Bias-aware Flow Attention and canonical attention pipeline.

- $\mathcal{L}_{FAPE}$: Main Frame Aligned Point Error Loss from the Structure Module block. This loss evaluates the predicted structure against ground truth 3D structure and scores all atoms in all backbone and side chain frames.

- $\mathcal{L}_{aux}$: Auxiliary Loss is also from the Structure Module. This term averages the FAPE and torsion losses of intermediate structures, which are iteratively generated by Structure Module blocks.

- $\mathcal{L}_{dist}$: It is an averaged cross-entropy loss for pair representation distogram prediction. It ensures that all entries in the pair representation have a clear relationship to the associated $(i, j)$ residue pair and assures that the pair representation will be useful for the structure module.

- $\mathcal{L}_{msa}$: It is an averaged cross-entropy loss for masked MSA prediction. During training, we added a Bert-like task by randomly masking some parts of MSA sequence tokens. This loss is intended to force the network to consider inter-sequence relationships, the co-evolution-like relationships, without explicitly encoding covariance statistics.

- $\mathcal{L}_{conf}$: Model confidence Loss scores the final predicted structure with per-residue $lDDT_{C_\alpha}$ against the ground truth structure.

### D.2 EVALUATION METRICS

We use the metrics below to comprehensively evaluate the memory usage of computation costs and the accuracy of predicted 3D protein structures.

**Computation cost.**

- Mem per block (GB): the memory usage for training each Liteformer/Evoformer block.
- Cost per block (ms): the time cost for training each Liteformer/Evoformer block.

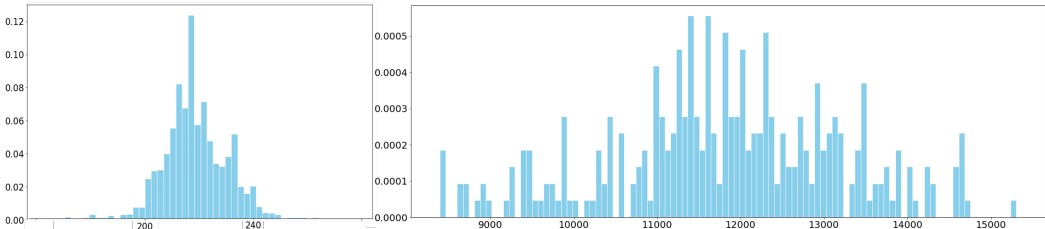

Figure 7: The distribution of protein single sequence length (left) and MSA sequence number (right). Single sequence length primarily ranges from 220 to 240, and its corresponding MSA sequence number mainly falls within the range of 11,000 to 13,000.

**Structure prediction.**

- TM-score (Yang & Jeffrey, 2004): Template modeling score (TM-score) is a useful complement to the fully automated assessment of protein structure predictions. This score evaluates all residue pairs to assess the quality of the predicted protein structure. The value of this metric ranges from 0 to 1, and shows a close coincidence with the results of the human-expert visual assessment.

- DockQ (Sankar & Björn, 2016): DockQ is a continuous protein-protein docking model quality measure with a single score in the range [0, 1]. This metric can estimate model quality in a more quantitative way, which is valuable for CASP community and protein docking field.

- RMSD: Root Mean Square Deviation (RMSD) is the most commonly used quantitative measure of the similarity between two structures' atomic coordinates. RMSD can be calculated for any type and subset of atoms. For Table **??**, we compare $C\alpha$ atoms of all residues.

- GDT-TS & GDT-HA (Zemla, 2003): Global Distance Test - Total Score (GDT-TS) is also used to quantify the similarity between a predicted protein structure and a reference structure. Instead of using the actual distance between $C\alpha$, GDT works with the percentage of $C\alpha$ that are found within certain cutoff distances of each other and is more robust against small fragment movements. GDT-TS gives an overall average measure of distance between predicted and reference amino acids, its values range from 0 (a meaningless prediction) to 100 (a perfect prediction). Global Distance Test - High Accuracy (GDT-HA) uses a shorter cutoff than GDT-TS and it is better at showing the proportion of $C\alpha$ with high prediction accuracy.

# E    MORE EXPERIMENTAL RESULTS

Figure 7 shows the distribution of single sequence length and MSA number. Figure 8 compares the single protein structure performance between Liteformer with ESM-2 150M (purple) and baseline Evoformer with ESM-2 150M (yellow). The x-axis is training global steps and the y-axis is the value of various metrics. We observe that the Liteformer can achieve comparable accuracy against Evoformer on CASP14, CASP15 and CAMEO.

Figure 6 shows the quantitative comparison with AlphaFold2 on single structure prediction. We randomly selected 10,000 general protein data as a training dataset and evaluated CASP14, CASP15 and CAMEO datasets. It is evident that the accuracy of structure predicted by Liteformer with AlphaFold2 outperforms Evoformer with AlphaFold2.

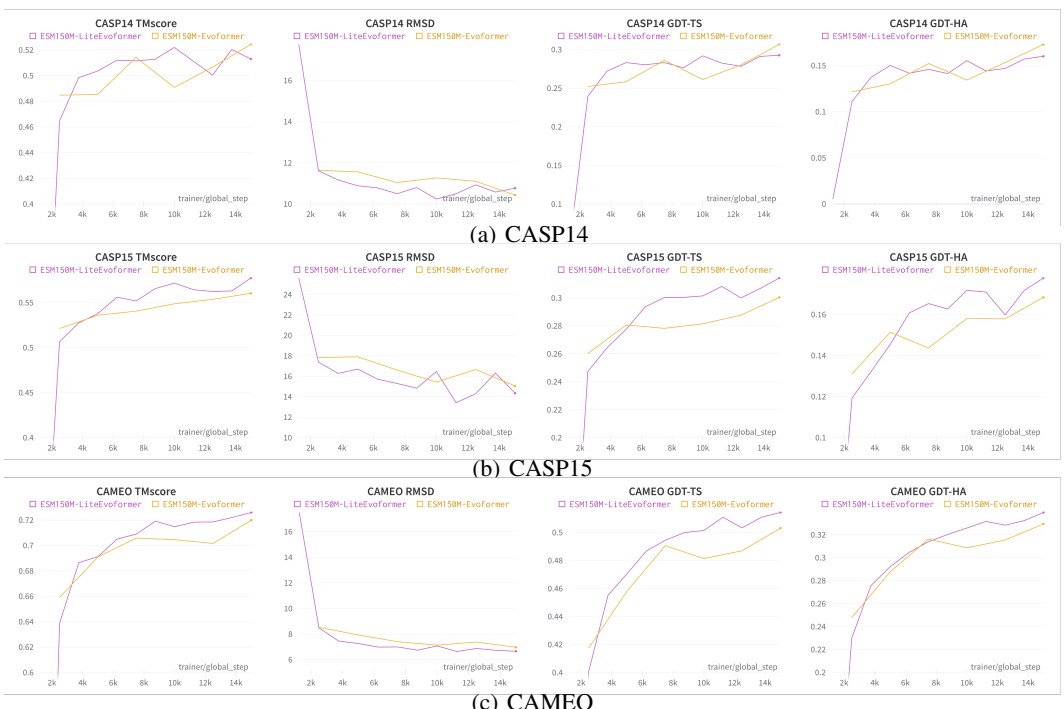

Figure 8: The performance comparison of Liteformer(purple) and Evoformer(yellow) with ESM-2 150M on CASP14,CASP15 and CAMEO in terms of TM-score, RMSD, GDT-TS and GDT-HA.

Table 6: Quantitative results comparison on CASP and CAMEO benchmarks.

z

| Dataset | Model w/ AF2 | TM-score ↑ | RMSD ↓ | GDT-TS ↑ | GDT-HA ↑ |
|---------|--------------|------------|--------|----------|----------|
| CASP14 | Evoformer | 0.5738 | 9.041 | 0.3808 | 0.2236 |
| | Liteformer | **0.6110** | **8.483** | **0.3877** | **0.2570** |
| CASP15 | Evoformer | 0.6102 | 15.588 | 0.3612 | 0.2105 |
| | Liteformer | **0.6269** | **15.220** | **0.3787** | **0.2176** |
| CAMEO | Evoformer | 0.7207 | 5.973 | 0.5537 | 0.3868 |
| | Liteformer | **0.7343** | **5.591** | **0.5867** | **0.3999** |

