# OpenReview forum: "Liteformer: Lightweight Evoformer for Protein Structure Prediction"
_ICLR.cc/2024/Conference — Submitted to ICLR 2024_

### Official Review · Reviewer_ffAR · 2023-10-20

**Soundness:** 1 poor
**Presentation:** 1 poor
**Contribution:** 2 fair
**Rating:** 3
**Confidence:** 4

**Summary:**

The authors introduce LiteFormer, a variant of the AlphaFold2 Evoformer inspired by Flowformer, a linear transformer. Compared to the original Evoformer, LiteFormer has lower complexity and is claimed to run faster and more memory-efficiently. The authors evaluate LiteFormer on monomeric and multimeric structure prediction.

**Strengths:**

It is true that Evoformer has high complexity, and I'm not yet aware of successful applications of the linear transformer literature to the AlphaFold2 architecture. It's also welcome that the authors evaluate on CASP and not just a small CAMEO dataset, like many other papers of this genre.

**Weaknesses:**

Some of the performance figures that motivate this entire paper seem questionable. It is claimed in Figure 1 that AlphaFold2 OOMs on sequences of length 800, e.g. If the authors are running inference using the same 80GB A100s they use later in the paper, this simply cannot be true; one can get away with longer sequences even on ColabFold (Mirdita et al., 2022), which runs on free-tier Google Colab GPUs. The authors of OpenFold were able to run an unmodified version of the original AlphaFold2 on sequences of length 2000 on a 40GB A100 (Ahdritz et al., 2022).

Separately, the choice of ESM-2 150M as a baseline for monomeric structure prediction is extremely confusing. Why not use unmodified AlphaFold2? Why not use a larger ESM-2 protein language model? Why mix and match these figures with AlphaFold2-Multimer evals? In general, details on the evaluation are very light (e.g. which CAMEO proteins were chosen? Why were all evaluation proteins filtered at 500? Which AlphaFold2 implementation served as the baseline?), to the point where it's difficult to know what is being run.

On top of that, the claimed performance improvements are comparable to or less significant than those of other optimized versions of AlphaFold2, none of which are mentioned here. FastFold, UniFold, OpenFold have all already improved AlphaFold2 with e.g. FlashAttention.

Misc.:

>However, since both the row and column dimensions of the pair representation, along with the row dimension of the MSA representation, are identical to the primary sequence length L.

>We trained 10,000 data for 5 days using 8 × DGX-A100 80G GPUs and inference on two multimeric datasets: VH-VL and DB5.5.

The paper is a bit sloppily written (see examples above). I'm not sure what the first sentence is saying. The second one contains no information about which 10,000 data points were used to train the model.

**Questions:**

>CASP14 & CASP15. The 14th & 15th Critical Assessment of protein Structure Prediction, from which we respectively select 42 and 48 single protein targets with sequence lengths less than 500.

Why filter in this way? The hardest proteins are the longest ones. Almost all entries at CASP15 did well on most of the proteins of length < 500 (excepting some orphans, etc.).

---

> ### Author Response · Authors · 2023-11-17
> **Response to Reviewer ffAR**
>
> Thank you for your feedback. We enhanced the draft's clarity and updated the experimental results for both Liteformer with ESM2-150M and Liteformer with unmodified AlphaFold2 on the complete CASP and CAMEO datasets. Finally, we explained the contrast between Flash Attention and our model. We provide detailed answers to your comments below.
>
> **It is claimed in Figure 1 that AlphaFold2 OOMs on sequences of length 800, e.g. If the authors are running inference using the same 80GB A100s they use later in the paper, this simply cannot be true; one can get away with longer sequences even on ColabFold (Mirdita et al., 2022), which runs on free-tier Google Colab GPUs. The authors of OpenFold were able to run an unmodified version of the original AlphaFold2 on sequences of length 2000 on a 40GB A100 (Ahdritz et al., 2022).**
>
> We appreciate the opportunity to clarify a crucial aspect that might have caused confusion. In Figure 1 and throughout the context, our focus is on comparing training lengths, as opposed to the inference lengths highlighted in the studies mentioned by the reviewer. It's essential to differentiate between training and inference processes. During training, the significant memory consumption predominantly arises from backward gradient propagation, thereby necessitating shorter sequence lengths compared to inference tasks. We would like to emphasize that our comparison specifically highlights training sequence lengths, and we've made this distinction explicit in the text and Figure 1's caption. Our intention was to address the challenges in memory and computational complexity during the training phase due to the substantial memory requirements for gradient backpropagation.
>
> **The choice of ESM-2 150M as a baseline for monomeric structure prediction is extremely confusing. Why not use unmodified AlphaFold2? Why not use a larger ESM-2 protein language model?**
>
> We would like to provide clarification regarding the selection of ESM2-2 150M as a baseline for monomeric structure prediction in our experiment. Liteformer is a versatile protein structure module that seamlessly integrates with both unmodified AlphaFold2 and other Protein Language Models, such as ESM-2. Our objective was to assess Lightformer's performance in the context of single protein structure prediction tasks.
>
> To further validate the effectiveness of our model, we included comparative experiments conducted with the unmodified AlphaFold2 framework. Due to the rebuttal's time limitation, we randomly select 10,000 general protein data from PDB and evaluate Liteformer and Evoformer on CASP and CAMEO datasets. Tab 6 shows that the accuracy predicted by Liteformer with AlphaFold surpasses Evoformer with AlphaFold on CASP and CAMEO.
>
> **Why mix and match these figures with AlphaFold2-Multimer evals?**
>
> Regarding the reviewer's concern about the integration of figures with AlphaFold2-Multimer evaluations, we would like to clarify that AlphaFold2-Multimer is tailored for complex structure prediction tasks. The assessment of its performance involves evaluating the accuracy of individual protein parts and docking performance. Both complex and single protein structure tasks necessitate assessing the predicted structure's accuracy using similar metrics. It's crucial to note that utilizing the same public metric doesn't imply a mix of results from two distinct tasks. Our experiment involved two tables explicitly demonstrating single and complex protein structure prediction performances separately, without amalgamating or analyzing them together. Figure 1 aims to show how our Liteformer achieves comparable results to the corresponding baseline model across different datasets. It's not about comparing two different tasks but rather demonstrating our model's performance across different datasets.
>
> **Which CAMEO proteins were chosen? Why were all evaluation proteins filtered at 500? Which AlphaFold2 implementation served as the baseline?**
>
> Thank you for your feedback. We inadvertently used a specific subset of data in our experiments due to an oversight in our experimental setup. Recognizing this error, we have re-conducted our experiments using the complete CASP and CAMEO datasets to ensure comprehensive and accurate results, as shown in Tab 2. It's important to note that during our initial experiments, a very small portion of excessively long-length data was filtered out. We apologize for any confusion caused by the previous use of filtering. The implementation of AlphaFold2 and AlphaFold-Multimer is from its original paper.

---

> ### Author Response · Authors · 2023-11-17
> **Response to Reviewer ffAR Part 2**
>
> **The claimed performance improvements are comparable to or less significant than those of other optimized versions of AlphaFold2, none of which are mentioned here. FastFold, UniFold, OpenFold have all already improved AlphaFold2 with e.g. FlashAttention.**
>
> Flash Attention is an implementation of the classical pairwise attention mechanism taking into account GPU details such as caches and communications. Flash Attention does not reduce the computational complexity in big $\mathrm{O}$ notation though. Such optimization is orthogonal to our work which focuses on deep learning architecture improvements to address computational complexity in big $\mathrm{O}$  notation. At the moment we did not find any public implementation of Flash Attention for Evoformer, especially the $O(L^3)$ component of the Evoformer.
>
> Furthermore, Flash Attention solely focuses on unbiased attention computation and lacks the capability to expedite computation with a predetermined bias. As the communication between MSA representation and Pair representation relies on biased attention, Flash Attention isn't directly applicable for acceleration within this context.
>
> Given these, we feel it will be unfair for us to invent a new Flash Attention implementation for Evoformer which is a new project on its own. In contrast, our contribution is to reduce the complexity of biased attention and achieve a similar accuracy for protein structure prediction tasks.
>
> That said, we thank the reviewer for bringing up Flash Attention in this context. We think implementation and design ideas from Flash Attention are orthogonal to our Liteformer deep learning architecture design, and it will be a good future direction to incorporate ideas from Flash Attention into Liteformer to further improve its efficiency.
>
> **"However, since both the row and column dimensions of the pair representation, along with the row dimension of the MSA representation, are identical to the primary sequence length L."  I'm not sure what the first sentence is saying. "We trained 10,000 data for 5 days using 8 × DGX-A100 80G GPUs" The second one contains no information about which 10,000 data points were used to train the model.**
>
> Thanks for your valuable suggestion. We have revised "However, since both the row and column dimensions of the pair representation, along with the row dimension of the MSA representation, are identical to the primary sequence length L." to "However, since both the row and column dimensions of the pair representation, along with the row dimension of the MSA representation, are the same as the primary sequence length L." and revised "We trained 10,000 data " to " We trained 10,000 randomly selected general protein data from PDB"
>
> **Table 2: Quantitative results comparison between Liteformer with ESM2-150M and Evoformer with ESM2-150M on CASP and CAMEO datasets.**
> | **Dataset** | **Model w/ ESM-2** | **TM-score** | **RMSD**   | **GDT-TS**  | **GDT-HA** |
> |-------------|--------------------|--------------|------------|-------------|------------|
> | CASP14      | Evoformer          | **0.4971**  | 10.086     | **0.2535** | **0.1347** |
> | CASP14      | Liteformer         | 0.4857       | **9.663**  | 0.2383      | 0.1220     |
> | CASP15      | Evoformer          | 0.5356       | 18.663     | 0.2433      | 0.1269     |
> | CASP15      | Liteformer         | **0.5517**   | **17.594** | **0.2536**  | **0.1377** |
> | CAMEO       | Evoformer          | 0.7112       | 5.962      | 0.4477      | 0.2770     |
> | CAMEO       | Liteformer         | **0.7177**   | **5.558**  | **0.4627**  | **0.2901** |
>
> **Table 6: Quantitative results comparison between Liteformer with AlphaFold2 and Evoformer with AlphaFold2 on CASP and CAMEO datasets.**
> | **Dataset** | **Model w/ AF2** | **TM-score** | **RMSD**   | **GDT-TS**  | **GDT-HA** |
> |-------------|--------------------|--------------|------------|-------------|------------|
> | CASP14      | Evoformer          | 0.5738  | 9.041   | 0.3808 | 0.2236 |
> | CASP14      | Liteformer         | **0.6110**      | **8.483**  |**0.3877**      | **0.2570**    |
> | CASP15      | Evoformer          | 0.6102     | 15.588    | 0.3612     | 0.2105    |
> | CASP15      | Liteformer         | **0.6269** | **15.220** | **0.3787** |**0.2176**|
> | CAMEO       | Evoformer          |0.7207 | 5.973 | 0.5537 | 0.3868  |
> | CAMEO       | Liteformer         | **0.7343**   | **5.591**  | **0.5867**  | **0.3999** |

---

> ### Author Response · Authors · 2023-11-20
> **Could you please let us know whether our responses and updated submission properly addressed your concern?**
>
> Dear Reviewer ffAR:
>
> Thank you very much for your time spent on our submission and your questions. We have tried to address your concerns in the response and updated submission--any feedback from you would be appreciated. If you have further comments, please kindly let us know. We hope for the opportunity to respond to them.
>
> Best wishes, Authors of paper 3633

---

> > ### Comment · Reviewer_ffAR · 2023-11-21
> >
> > My apologies for the delayed response.
> >
> > > Tab 6 shows that the accuracy predicted by Liteformer with AlphaFold surpasses Evoformer with AlphaFold on CASP and CAMEO.
> >
> > Thank you for including this information. Unfortunately, while I recognize the time and computational constraints involved, the Evoformer you use here seems very undertrained. Doing some back of the envelope calculations, w/ training iteration times of ~8 seconds and AlphaFold's original batch size of 120 (implying 120 / 8 = 15-way gradient accumulation), training for 5 days gets you about 3.5k full training steps, which isn't enough for the model to reach reasonably high accuracies. The Evoformer in Tables 2 and 6 is performing worse than AlphaFold1, and so I don't think this is a very meaningful comparison.
> >
> > > Our objective was to assess Lightformer's performance in the context of single protein structure prediction tasks.
> >
> > This doesn't answer my question why you didn't use a larger ESM model.
> >
> > > We think implementation and design ideas from Flash Attention are orthogonal to our Liteformer deep learning architecture design, and it will be a good future direction to incorporate ideas from Flash Attention into Liteformer to further improve its efficiency.
> >
> > FlashAttention is just one example of the performance improvements implemented by papers like FastFold, UniFold, OpenFold, etc. While I appreciate that your contribution reduces the computational complexity of the operations, I think it's still very important for you to compare to other efficient versions of AlphaFold.
> >
> > You can consider concerns I didn't bring up here to be resolved.

---

> > > ### Author Response · Authors · 2023-11-22
> > > **Response to Reviewer ffAR**
> > >
> > > Thank you for your valuable feedback on our paper. Our detailed response to your question is as below:
> > >
> > > **The Evoformer you use here seems very undertrained. The Evoformer in Tables 2 and 6 is performing worse than AlphaFold1, and so I don't think this is a very meaningful comparison.**
> > >
> > > Regarding Table 2, we employed the Evoformer with the ESM2 framework rather than directly utilizing the AlphaFold2 model. Thus, it's inappropriate to compare our training steps directly to AlphaFold2 and label our training as undertrained. Furthermore, in the appendix, we included training curves for Evoformer with ESM2, clearly demonstrating the convergence during the training process. In numerous studies that utilize protein language models (PLMs) instead of multiple sequence alignments (MSA), such as OmegaFold and ESMFold, it is observed that the structural prediction accuracy using PLMs tends to be lower than models like AlphaFold2 that leverage MSA. We attribute this to the fact that PLMs might not capture the rich biological information present in MSA during pre-training. Regarding Table 6, due to time constraints in the rebuttal phase and limitations in GPU computational resources, we only utilized 10,000 training data. Admittedly, our accuracy may not match that of AlphaFold2, which utilized 10 million training data. However, the main focus of Table 2 and Table 6 is to demonstrate that our model, despite reduced complexity, manages to maintain considerable performance without significant loss.
> > >
> > > **This doesn't answer my question why you didn't use a larger ESM model**
> > >
> > > As previously highlighted, our main focus was not specifically on the model variant but rather on evaluating improvements within the framework for single-structure prediction.
> > >
> > > While larger ESM models might indeed offer potential improvements, the primary constraint we faced was computational resources. Larger models demand significantly more computational power and prolonged training times. Our decision was influenced by the practical limitations in computing resources rather than a deliberate disregard for larger ESM architectures' potential benefits.
> > >
> > > **FlashAttention is just one example of the performance improvements implemented by papers like FastFold, UniFold, OpenFold, etc. While I appreciate that your contribution reduces the computational complexity of the operations, I think it's still very important for you to compare to other efficient versions of AlphaFold.**
> > >
> > > The nature of our work lies in algorithmic optimization for complexity reduction, which, by design, follows a different trajectory compared to memory or communication optimizations in papers Reviewer ffAR mentioned. While we acknowledge the importance of those domains, comparing them directly would disregard the fundamental disparities in the optimization goals and methodologies.
> > >
> > > We appreciate the reviewer's inclusion of these papers in this discussion. We believe that the implementation and design concepts presented in these papers stand orthogonal to our Liteformer deep learning architecture. Incorporating memory and communication optimization ideas into Liteformer could serve as a promising future avenue to enhance its efficiency.

---

### Official Review · Reviewer_WrTu · 2023-10-30

**Soundness:** 2 fair
**Presentation:** 2 fair
**Contribution:** 2 fair
**Rating:** 5
**Confidence:** 3

**Summary:**

This paper proposes Liteformer, a novel and lightweight variant of  Evoformer used in AlphaFold2 for protein structure prediction. Liteformer introduces a new mechanism called Bias-aware Flow Attention (BFA), which linearizes the biased attention for third-order tensors, such as multiple sequence alignment (MSA) and pair representation, with O(L) complexity.  Liteformer reduces the memory consumption and training speed of Evoformer by up to 44* and 23%,  respectively while maintaining competitive accuracy on various protein structure benchmarks.

**Strengths:**

- The paper proposes a novel and efficient variant of Evoformer, the core module of AlphaFold2, which is a state-of-the-art model for protein structure prediction.
- The paper introduces a new mechanism called Bias-aware Flow Attention, which linearizes the biased attention for third-order tensors with O(L) complexity, while preserving the evolutionary and geometric information from MSA and pair representations1.
- The paper demonstrates the effectiveness and efficiency of Liteformer on various protein structure benchmarks, such as CASP14, CASP15, CAMEO, VH-VL, and DB5.5, showing that it can reduce the memory consumption and training speed of Evoformer by up to 44% and 23%, respectively, while maintaining competitive accuracy.

**Weaknesses:**

- Unclear motivation: The author emphasizes the huge computational complexity reduction brought by converting attention module  into a flow network. However, how this method affects the computational complexity is only mentioned at Eeq 7 in Section 3. I think there should be a separate paragraph in the introduction detailing the motivations for using flow network.
- The claim does not correspond to the experimental results. In Sec 2 and 3, the authors mainly claim that BFA can reduce computational complexity. However, the experimental results show that memory usage is the main advantage of BFA during training. The drop in computation time is less pronounced. The authors in sec 4 simply boil this down to reduced graph overhead when backpropagated. In my opinion, it is not helpful to prove the validity of this method.

**Questions:**

Please elaborate on the motivation for using streaming networks and how it leads to a reduction in computational complexity.

---

> ### Author Response · Authors · 2023-11-17
> **Response to Reviewer WrTu**
>
> Thank you for your thorough assessment and comments on our manuscript.  We are excited that you appreciated the contribution and the novelty of our work. We have enhanced the clarity of our draft by indicating our motivation for using flow network in the Introduction section and revising the complexity analysis of our model in Section 2 and Section 3. We respond to your comments below.
>
> **I think there should be a separate paragraph in the introduction detailing the motivations for using flow network.**
>
> We have added a paragraph in the Introduction section to illustrate the motivation for utilizing flow network as below: " The diversity of protein sequences requires non-trivial attention to generality. However, previous linear attention mechanisms mainly use similarity decomposition to approximate Softmax($\mathrm{QK^T}$) with different kernel functions, which reintroduce inductive bias and devise linear complexity at the expense of generality. Therefore, we turn to reconstructing the biased attention mechanism from a new perspective, a flow network perspective. "
>
> **The claim does not correspond to the experimental results. In Sec 2 and 3, the authors mainly claim that BFA can reduce computational complexity at Equation 7. However, the experimental results show that memory usage is the main advantage of BFA during training.**
>
> We'd like to address the point you've raised, as we suspect there might be some misunderstanding. Our analysis in Section 2 and Section 3 have extensively delved into both memory and computational complexities within the manuscript, aligning closely with the experimental outcomes. In the context following Equations 1-10, we have intentionally provided insights into the tensor shapes involved in each operation, and we've analyzed the computational and memory complexities associated with each operation, albeit using the term "complexity." Following your advice, we updated the analysis in Section 2 and Section 3 using the term "memory and computational complexity".

---

> > ### Comment · Reviewer_WrTu · 2023-12-01
> >
> > Thank you for your detailed response. However, after considering the issues raised by other reviewers and your responses, especially the comparison with Liteformer, I have decided to maintain the original score.

---

> ### Author Response · Authors · 2023-11-20
> **Could you please let us know whether our responses and updated submission properly addressed your concern?**
>
> Dear Reviewer WrTu:
>
> Thank you very much for your time spent on our submission and your questions. We have tried to address your concerns in the response and updated submission--any feedback from you would be appreciated. If you have further comments, please kindly let us know. We hope for the opportunity to respond to them.
>
> Best wishes, Authors of paper 3633

---

> ### Author Response · Authors · 2023-11-21
> **Could you please let us know whether our responses and updated submission properly addressed your concern?**
>
> Dear Reviewer WrTu,
>
> We extend our sincere gratitude for investing your time in reviewing our submission and raising pertinent questions. Our response aims to tackle your concerns, reflected in the updated submission. If you have further comments, please kindly let us know. We hope for the opportunity to respond to them.
>
> Best regards, Authors of paper 3633

---

> ### Comment · Area_Chair_CKua · 2023-11-22
> **Replies to author comments**
>
> Dear reviewer,
>
> Thank you very much for your work evaluating this review.
>
> It is critical that you urgently address the author's responses, acknowledge their response, and eventually adjust your rating if warranted.
>
> Best,
>
> AC

---

### Official Review · Reviewer_9JiD · 2023-11-01

**Soundness:** 3 good
**Presentation:** 3 good
**Contribution:** 2 fair
**Rating:** 5
**Confidence:** 4

**Summary:**

In this paper, the authors raise the problem on the high memory and time cost of the Evoformer module
used in AlphaFold2.
To solve this, they propose a lightweight variant of Evoformer named Liteformer. Through a bias-aware flow
attention (BFA) mechanism, the complexity of Liteformer is reduced to a lower quantity, compared with
original Evoformer.
Extensive experiments show the great effectiveness of the proposed method in terms of both memory
occupation and training acceleration, while keeping the competitive accuracy in protein structure
prediction.

**Strengths:**

1. The propose mechanism is very interesting and useful for the development of applications in recent
years.
2. Existing frameworks equipped with the proposed BFA modules can achieve competitive performance while
reducing the time of training and inference a lot.

**Weaknesses:**

1. In Fig. 2, the pipeline of the blocks in Evoformer and Liteformer is exactly the same. It's better to be combined into one figure. The difference between BFA and original attention is the key to which
should be compared like this.
2. The proposed method shows a great improvement in efficiency and memory cost. But there have
been some more general works that play a similar role, such as memory-efficient attention and flash
attention [Ref_1]. Can they jointly improve the training of the network? More ablation studies between
them should be provided for further comparison.
3. The proposed BFA module seems more likely to be a general attention module, instead of a theme-related
(protein-related) approach, which decreases the significance.
[Ref_1]. Dao, Tri, et al. "Flashattention: Fast and memory-efficient exact attention with io-awareness."
Advances in Neural Information Processing Systems 35 (2022): 16344-16359.

**Questions:**

What's the principle of selecting the targets from the CASP14, CASP15, CAMEO, and VH-VL, and why
sequence lengths of them are restricted to be shorter than 500, since Liteformer has a better ability to
handle the longer sequences?

---

> ### Author Response · Authors · 2023-11-17
> **Response to Reviewer 9JiD**
>
> Dear Reviewer,
>
> Thank you for your thorough review. We are glad that you appreciate the contribution and novelty of our work. We improved the clarity of our draft, updated the method's figures, and indicated our model's tailoredness to protein tasks. Finally, we re-conducted the monomeric protein structure prediction on the complete CASP and CAMEO datasets and explained the contrast between Flash Attention and our model. We appreciate your insightful suggestions and provide detailed answers to your comments below.
>
> **In Fig. 2, the pipeline of the blocks in Evoformer and Liteformer is exactly the same. It's better to be combined into one figure. The difference between BFA and original attention is the key to which should be compared like this.**
>
> Following your advice, we have updated the Liteformer architecture figure in Figure 2 and the comparison between BFA and canonical attention in Figure 6. Figure 2 shows the distinction between Evoformer and our Liteformer architectures within a single panel. Through the implementation of Bias-aware Flow Attention, our Liteformer notably diminishes the memory and computational complexities from quadratic to linear in MSA row-wise updates and from cubic to quadratic in pair-wise triangle updates. Additionally, we have incorporated the attention computation process alongside our Bias-aware Flow Attention in Figure 6 to aid in contrasting our linear attention algorithm with the canonical attention method.
>
> **More ablation studies between Liteformer and Flash Attention should be provided for further comparison. Can they jointly improve the training of the network?**
>
> Flash Attention is an implementation of the classical pairwise attention mechanism taking into account GPU details such as caches and communications. Flash Attention does not reduce the computational complexity in big $\mathrm{O}$ notation though. Such optimization is orthogonal to our work which focuses on deep learning architecture improvements to address computational complexity in big $\mathrm{O}$  notation. At the moment we did not find any public implementation of Flash Attention for Evoformer, especially the $O(L^3)$ component of the Evoformer.
>
> Furthermore, Flash Attention solely focuses on unbiased attention computation and lacks the capability to expedite computation with a predetermined bias. As the communication between MSA representation and Pair representation relies on biased attention, Flash Attention isn't directly applicable for acceleration within this context.
>
> Given these, we feel it will be unfair for us to invent a new Flash Attention implementation for Evoformer which is a new project on its own. In contrast, our contribution is to reduce the complexity of biased attention and achieve a similar accuracy for protein structure prediction tasks.
>
> That said, we thank the reviewer for bringing up Flash Attention in this context. We think implementation and design ideas from Flash Attention are orthogonal to our Liteformer deep learning architecture design, and it will be a good future direction to incorporate ideas from Flash Attention into Liteformer to further improve its efficiency.
>
> **The proposed BFA module seems more likely to be a general attention module, instead of a theme-related (protein-related) approach, which decreases the significance.**
>
> Regarding the perception that our approach is more of a general algorithm rather than tailored to a specific task, we would like to clarify that linearization of biased attention within our BFA method directly addresses the unique demands of protein modeling. In protein modeling, the exchange of information between MSA and Pair-wise representations crucially relies on biased attention mechanisms. Our focus on linearizing biased attention specifically caters to the requirements of this domain, ensuring an optimal strategy for enhancing the efficiency of protein structure prediction tasks.
>
> **What's the principle of selecting the targets from the CASP14, CASP15, CAMEO, and VH-VL, and why sequence lengths of them are restricted to be shorter than 500, since Liteformer has a better ability to handle the longer sequences?**
>
> Thank you for your feedback. We inadvertently used a specific subset of data in our experiments due to an oversight in our experimental setup. Recognizing this error, we have updated our experimental results using the entire CASP and CAMEO dataset to ensure comprehensive and accurate results, as shown in Tab 2 in the main context. It's important to note that during our initial experiments, a very small portion of excessively long-length data was filtered out. We apologize for any confusion caused by the previous use of filtering.

---

> ### Author Response · Authors · 2023-11-17
> **Response to Reviewer 9JiD Part 2**
>
> **Table 2: Quantitative results comparison between Liteformer with ESM2-150M and Evoformer with ESM2-150M on CASP and CAMEO datasets.**
> | **Dataset** | **Model w/ ESM-2** | **TM-score** | **RMSD**   | **GDT-TS**  | **GDT-HA** |
> |-------------|--------------------|--------------|------------|-------------|------------|
> | CASP14      | Evoformer          | **0.4971**  | 10.086     | **0.2535** | **0.1347** |
> | CASP14      | Liteformer         | 0.4857       | **9.663**  | 0.2383      | 0.1220     |
> | CASP15      | Evoformer          | 0.5356       | 18.663     | 0.2433      | 0.1269     |
> | CASP15      | Liteformer         | **0.5517**   | **17.594** | **0.2536**  | **0.1377** |
> | CAMEO       | Evoformer          | 0.7112       | 5.962      | 0.4477      | 0.2770     |
> | CAMEO       | Liteformer         | **0.7177**   | **5.558**  | **0.4627**  | **0.2901** |

---

> ### Author Response · Authors · 2023-11-20
> **Could you please let us know whether our responses and updated submission properly addressed your concern?**
>
> Dear Reviewer 9JiD:
>
> Thank you very much for your time spent on our submission and your questions. We have tried to address your concerns in the response and updated submission--any feedback from you would be appreciated. If you have further comments, please kindly let us know. We hope for the opportunity to respond to them.
>
> Best wishes, Authors of paper 3633

---

> ### Author Response · Authors · 2023-11-21
> **Could you please let us know whether our responses and updated submission properly addressed your concern?**
>
> Dear Reviewer 9JiD,
>
> We extend our sincere gratitude for investing your time in reviewing our submission and raising pertinent questions. Our response aims to tackle your concerns, reflected in the updated submission. If you have further comments, please kindly let us know. We hope for the opportunity to respond to them.
>
> Best regards, Authors of paper 3633

---

> ### Comment · Area_Chair_CKua · 2023-11-22
> **Please address author response**
>
> Dear reviewer,
>
> Thank you very much for your work evaluating this review.
>
> It is critical that you urgently address the author's responses, acknowledge their response, and eventually adjust your rating if warranted.
>
> Best,
>
> AC

---

### Meta-Review · Area_Chair_CKua · 2023-12-05

**Metareview:**

The reviewers brought up some important issues including having to do with novelty, and gaps between claims and experimental results. The authors energetically made their case, but ultimately did not convince the reviewers.

**Justification For Why Not Higher Score:**

Based on my own evaluation - and the reviewers' - there are too many issues for this work to be accepted at ICLR

**Justification For Why Not Lower Score:**

N?A

---

### Decision · Program_Chairs · 2024-01-16

Reject